# Anaesthesia Choice for Creation of Arteriovenous Fistula (ACCess) study protocol : a randomised controlled trial comparing primary unassisted patency at 1 year of primary arteriovenous fistulae created under regional compared to local anaesthesia

Alan JR Macfarlane [1,2] Rachel J Kearns,[1,2] Marc James Clancy,[3,4] David Kingsmore,[4,5] Karen Stevenson,[3] Andrew Jackson,[3] Patrick Mark,[4,6] Margaret Aitken,[3] Ramani Moonesinghe,[7,8] Cecilia Vindrola-Padros [9] Lucian Gaianu,[10] Gavin Pettigrew,[11,12] Reza Motallebzadeh,[13,14] Nikolaos Karydis,[15] Alex Vesey,[16] Rita Singh,[17] Thalakunte Muniraju,[18] Stuart Suttie,[19] Alex McConnachie [20] Kirsty Wetherall,[20] Kariem El-Boghdadly [21,22] Rosemary Hogg,[23] Iain Thomson,[24] Vishal Nangalia,[25] Emma Aitken,[3,4] The Access collaborative group

For numbered affiliations see end of article.

**Correspondence to**
Professor Alan JR Macfarlane;
alan.macfarlane@glasgow.ac.uk

## ABSTRACT

**Introduction** Arteriovenous fistulae (AVF) are the 'gold standard' vascular access for haemodialysis. Universal usage is limited, however, by a high early failure rate. Several small, single-centre studies have demonstrated better early patency rates for AVF created under regional anaesthesia (RA) compared with local anaesthesia (LA). The mechanistic hypothesis is that the sympathetic blockade associated with RA causes vasodilatation and increased blood flow through the new AVF. Despite this, considerable variation in practice exists in the UK. A high-quality, adequately powered, multicentre randomised controlled trial (RCT) is required to definitively inform practice.

**Methods and analysis** The Anaesthesia Choice for Creation of Arteriovenous Fistula (ACCess) study is a multicentre, observer-blinded RCT comparing primary radiocephalic/brachiocephalic AVF created under regional versus LA. The primary outcome is primary unassisted AVF patency at 1 year. Access-specific (eg, stenosis/thrombosis), patient-specific (including health-related quality of life) and safety secondary outcomes will be evaluated. Health economic analysis will also be undertaken.

**Ethics and dissemination** The ACCess study has been approved by the West of Scotland Research and ethics committee number 3 (20/WS/0178). Results will be published in open-access peer-reviewed journals within 12 months of completion of the trial. We will also present our findings at key national and international renal and anaesthetic meetings, and support dissemination of trial outcomes via renal patient groups.

### Strengths and limitations of this study

► This is a prospective, multicentre, randomised, observer-blinded trial designed to examine whether primary radiocephalic/brachiocephalic arteriovenous fistulae created under regional anaesthesia (RA) rather than local anaesthesia (LA) have better 1-year primary unassisted functional patency.

► With 566 participants, this will be the largest trial to date comparing RA to LA and will address criticisms of previous smaller, single-centre randomised trials.

► An associated cost-effectiveness analysis will provide sufficient evidence to guide practice and policy.

► The main limitation of this study is that in predialysis patients, the primary endpoint uses surrogate markers of fistula patency (clinical assessment and ultrasound scan criteria) rather than successful use of the fistula for haemodialysis.

**Trial registration number** ISRCTN14153938.
**Sponsor** NHS Greater Glasgow and Clyde GN19RE456, Protocol V.1.3 (8 May 2021), REC/IRAS ID: 290482.

## INTRODUCTION

The incidence of kidney failure requiring kidney replacement therapy has increased substantially over the last 30 years, and over 25 000 people in the UK are currently treated with maintenance haemodialysis (HD).[1]

Kidney disease has a significant impact on both longevity and quality of life and places considerable demand on healthcare resources.[2] Vascular access is 'a major modifiable risk factor' in terms of patient experience and outcome on HD, with arteriovenous fistulae (AVF) being the preferred mode of vascular access.[3] Patients dialysing via AVF experience less infective and thrombotic complications, and are three times less likely to be admitted to the hospital than their counterparts with central venous catheters (CVCs).[3] Such frequent hospitalisations have a negative impact on health-related quality of life (HRQoL).[4] Furthermore, there are observational data demonstrating superior survival in patients successfully dialysing via AVF compared with those using CVCs or prosthetic arteriovenous grafts (AVGs) for dialysis.[5]

Despite these benefits, universal adoption of AVF remains suboptimal. The most recent UK Renal Registry (UKRR) Multisite Dialysis Access Audit highlighted that nearly 80% of dialysis units in England, Wales and Northern Ireland still fail to achieve renal association targets, which recommend that 60% of incident patients receive HD via AVF or AVGs.[6 7] One principal obstacle to the widespread use of AVF is 'failure to mature', with early failure rates approaching 50%.[8 9] Any intervention that improved AVF maturation should confer significant benefit to patient health and well-being, reduce surgical workload and deliver cost savings.

Anaesthetic technique is one such factor believed to influence AVF maturation and outcome. Regional anaesthesia (RA), unlike local anaesthesia (LA), generates a sympathetic blockade. The mechanistic hypothesis is this sympathetic blockade results in vasodilatation, improved tissue oxygenation and increased blood flow through the new fistula, therefore reducing early thrombosis.[8 10] Several studies have demonstrated superior short-term patency rates of AVF created under brachial plexus block (BPB) compared with LA.[8 11] In the only randomised controlled trial (RCT) to date with prolonged follow-up, RA improved both early and 1-year functional AVF patency compared with LA.[12] A concomitant health economic analysis using HRQoL data extrapolated from the literature established net cost savings at 1 year and an incremental cost-effectiveness ratio (ICER) of approximately £12 900 per quality-adjusted life year (QALY) gained over a 5-year time horizon with RA.

Both European Society for Vascular Surgery and European Renal Association guidelines suggest considering using RA for all primary AVF.[13 14] The Kidney Disease Outcomes Quality Initiative Vascular Access guidelines disagree, stating choice of anaesthesia should be based on operator discretion.[15] Such disparity regarding the choice of anaesthesia for AVF creation is reflected across UK centres, with significant variation in practice.[16] While this is in part due to a lack of anaesthetic availability or capability, the failure to modify local practices perhaps also reflects the lack of strong evidence. All RCTs to date have been single-centre, with some suffering from methodological flaws.[11] Further, more robust, health

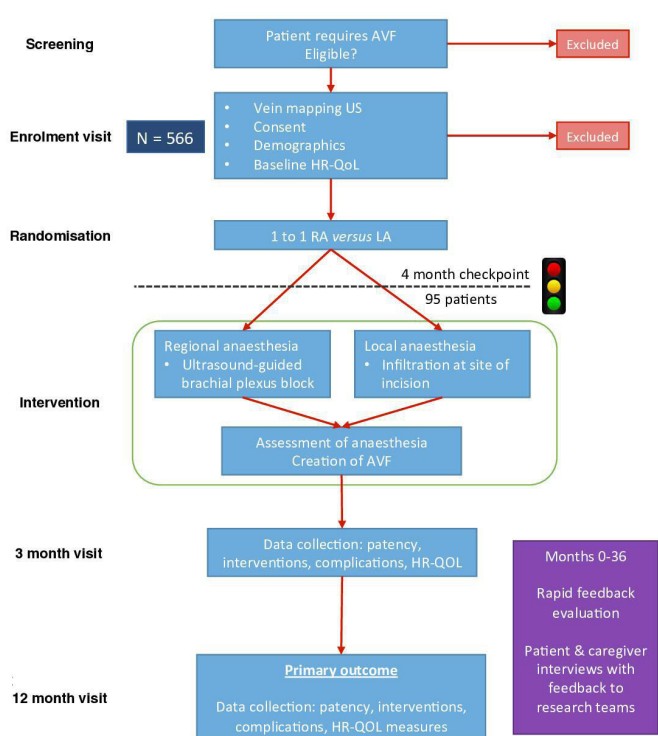

**Figure 1** Participant timeline. AVF, arteriovenous fistula; HRQoL, health-related quality of life; LA, local anaesthesia; RA, regional anaesthesia; US, ultrasound.

economics analysis is required to establish whether any potential durable clinical benefit could be offset against the longer procedural times, need for a skilled anaesthetist and the additional upfront costs associated with RA. Only a definitive, adequately powered, multicentre RCT with associated cost-effectiveness analysis will provide sufficient evidence to change practice and policy. The Anaesthesia Choice for Creation of Arteriovenous Fistula (ACCess) trial aims to address this issue, with the primary objective being to determine whether or not the sympathetic blockade associated with regional compared with LA translates into clinical improvements in long-term functional fistula patency.

## METHODS AND ANALYSIS
The ACCess study is a multicentre, observer-blinded, parallel group, superiority RCT with an internal pilot and embedded process evaluation study comparing patients undergoing primary radiocephalic fistula (RCF) or brachiocephalic fistula (BCF) creation under RA versus LA. The primary outcome is unassisted functional AVF patency at 1 year. The participant timeline is outlined in figure 1.

## Participants
Patients will be recruited from high-volume (>150 cases per year) and medium-volume (>50 cases per year) UK centres providing vascular access for HD.

Box 1   Inclusion and exclusion criteria

**Inclusion criteria**
► All adult patients (≥18 years old) with kidney failure requiring kidney replacement therapy or chronic kidney disease stage IV or V referred for primary radiocephalic fistula or brachiocephalic fistula creation.

**Exclusion criteria**

General
► Patients unable or unwilling to provide informed consent.
► Patient preference for general or alternative anaesthesia.
► Active infection at surgical or anaesthetic site.

Access specific
► Previous ipsilateral AVF creation (a previous attempt at distal AVF creation which fails immediately is not considered a contraindication, but any distal access which has previously run sufficiently to mature the outflow vein or proximal revision of an existing AVF is considered a contraindication).
► Known ipsilateral cephalic arch or central venous stenosis (even if previously treated).
► USS evidence of stenosis in inflow artery.
► Radial or brachial artery of <1.8 mm diameter and/or cephalic vein of <2 mm at the wrist or <3 mm at the elbow (with tourniquet) on preoperative USS.[29]

Contraindications to anaesthetic agents/ technique
► Allergy to local anaesthesia or any excipient agents.
► Acquired or inherited coagulopathy (including warfarin/heparin/novel oral anticoagulant use where it has not been possible to stop the anticoagulation in anticipation of surgery) and/or platelets of <75 or International Normalised Ratio (INR) of >1.4.[30]
► Significant pre-existing neurological disorder affecting the upper limb.
► Weight of <45 kg.

AVF, arteriovenous fistula; USS, ultrasound scan.

## Recruitment

Potentially eligible participants will be identified from vascular access and/or 'predialysis' (low-clearance) clinics and theatre waiting lists by the clinical team. A vein mapping ultrasound (US) will be performed to ensure minimum vessel characteristics. We anticipate 12–20 centres recruiting an average of two patients/centre/month. Recruitment is due to commence May 2021 and is anticipated to take 2 years. Preliminary results are expected late 2024.

## Inclusion criteria

Inclusion criteria are outlined in box 1.

## Exclusion criteria

Exclusion criteria are outlined in box 1.

## Allocation

A central randomisation facility (interactive web response system) at the Robertson Centre for Biostatistics (RCB), University of Glasgow, will randomise patients 1:1 to the intervention group (RA) or the comparator group (LA). The randomisation list will be created by a computer-generated program using a method of permuted blocks stratified by centre, dialysis status (predialysis/HD) and site of AVF (RCF/BCF). Randomisation will take place at a patient, not centre, level to minimise bias from variation in surgical and dialysis practice. The randomisation list, the program that generated it and the random seed used will be stored in a secure network location, accessible only to those responsible for provision of the randomisation system. Clinicians responsible for delivering the perioperative care will perform the web-based randomisation.

## Intervention

The choice of anaesthetic agents is influenced largely by the successful use of these combinations in a previous study,[8] the ready availability of these drugs within the UK, acceptability to collaborating centres, and the ability of the combination to provide both rapid onset and prolonged duration of block.[17]

### Interventional arm: RA

An US-guided supraclavicular or axillary BPB will be administered by a consultant anaesthetist trained in RA or a trainee practising under direct supervision. The supraclavicular approach will be considered first-line, unless the anatomy or patient risk profile is unfavourable. In patients on antiplatelets or other anticoagulants, the choice between supraclavicular and axillary block will be at the anaesthetist's discretion, taking into account 'compressibility, vascularity and consequences of bleeding'.[18] Where pulmonary disease is present, an axillary nerve block eliminates the risk of pneumothorax or temporary phrenic nerve paralysis.

A 1:1 mixture of 0.5% L-bupivacaine and 1% lidocaine, mixed with epinephrine to 1 in 400 000 final concentration, will be used (online supplemental appendix 1). Maximum dose limits are 2 mg/kg for bupivacaine and 7 mg/kg for lidocaine with epinephrine, recognising the effects are additive. The volume of LA injected must account for patient weight and maximum dose limits while considering the need for LA supplementation. In a study where the median patient weight was 66 kg, the $ED_{95}$ (effective volume in 95% of patients) for supraclavicular blocks was 27 mL.[19] A minimum volume of 25 mL must be injected for patients over 60 kg. This is reduced to 20 mL for patients weighing 51–60 kg and 15 mL in patients weighing <45 kg (online supplemental appendix 1). A suggested supraclavicular technique involves depositing a minimum of 25% of LA in the 'corner pocket' between the first rib and the subclavian artery and the remainder posterolateral to the plexus, avoiding deliberate intra-cluster injection.[20] For axillary blocks, the same minimum volumes must be used, targeting 25% of the LA to the musculocutaneous nerve, with the remainder deposited around the ulnar, median and radial nerves, as well as the cutaneous nerves of the arm and forearm if visualised.

Sensory and motor block of musculocutaneous, median, radial and ulnar nerves will be recorded every 5 min using a validated 3-point scale.[21] Sensory blockade of the medial cutaneous nerve of the forearm and arm will also be recorded. Measurements will be continued until

either sensory block is adequate or 30 min has elapsed, at which point the block may be supplemented by targeted US-guided axillary or midhumeral injection as appropriate.

### Comparator arm: LA

A 1:1 mixture of 0.5% L-bupivacaine and 1% lidocaine will be infiltrated around the operative site by the operating surgeon. After 5 min, adequacy of anaesthesia will be tested by application of a painful stimulus and additional LA infiltration administered as required. Maximum dose limits of 2 mg/kg for bupivacaine and 3 mg/kg for lidocaine will be observed, recognising the effects are additive.

### Management of a 'failed block' (or failure of LA)

A failed block will be defined as any block that, despite the targeted intervention described previously, requires additional supplementation with LA, analgesia, conversion to general anaesthesia or abandonment of surgery. The algorithm for failed block or 'failed LA' will be as follows:

1. Supplementation at surgical site with LA (1% lidocaine) up to maximum cumulative LA dosage.
2. Intravenous sedation and analgesia at the discretion of the anaesthetist.
3. General anaesthesia.
4. Abandonment of procedure: decision to be made following discussion between operating surgeon and anaesthetist if deemed unsafe to proceed with general anaesthesia.

The Trial Steering Committee (TSC)/Independent Data Monitoring Committee (IDMC) will monitor the number of failed blocks for patient safety and quality assurance throughout the study.

### Fistula surgery

A standard approach to the vessels will be via a transverse incision at, or just below, the elbow crease for BCF and longitudinal or curvilinear incision at the wrist for RCF. The cephalic vein (or median cubital vein if suitable) will be dissected and skeletalised for a short length proximally and distally. Visible branches will ligated and divided. The vein will be divided, spatulated where appropriate and flushed with heparinised saline. The artery will be dissected and controlled with bulldog clamps or slings. The decision to use median cubital, perforating branch or true outflow cephalic vein for the anastomosis will be at the surgeon's discretion, as will be the decision to create a proximal radial or ulnar–cephalic fistula at the elbow. The size of the arteriotomy will be based on individual patient risks and benefits, but arteriotomies will generally be between 3 and 5 mm in length on the brachial artery and 7–10 mm on the radial artery. An end-to-side anastomosis of vein to artery will be performed with continuous 6.0 (elbow) or 7.0 (wrist) prolene.

### Blinding

Due to the systemic effects of RA (motor blockade, visible venodilatation, etc), which do not occur with LA, it will not be possible to blind the patients, surgical or anaesthetic teams. Dialysis staff and staff performing follow-up visits will be blinded to the intervention. Ultrasound scan (USS) will also provide independent objective assessment of the AVF. The statisticians and health economist will be blinded to the intervention.

### Outcomes

The primary outcome is unassisted functional AVF patency at 1 year, defined as the ability of the AVF to uninterruptedly deliver the prescribed dialysis without intervention.[22] In predialysis patients, this will be assessed both clinically by an experienced, blinded dialysis nurse and ultrasonographically, the target being 4 mm diameter and access flow being >500 mL/min.[23]

All secondary outcome measures will be assessed at 3 and 12 months. These were chosen with two considerations: patient-centred care and to facilitate health economic analysis. The secondary outcomes reflect the 'standard CKD set' recommended by the International Consortium for Health Outcomes Measurement (ICHOM) CKD Working Group.[24] Key safety and efficacy outcomes for US-guided regional nerve blocks outlined by the National Institute for Health and Clinical Excellence will be recorded.[25] Secondary outcome measures are listed in box 2. These include access-specific outcomes (eg, reinterventions); patient-specific outcomes (eg, mortality); HRQoL outcomes (eg, KD-QOL); quality and speed of onset of anaesthesia; and safety outcomes.

### Economic evaluation

A cost-effectiveness analysis will be conducted alongside the clinical trial. Two complementary cost-effectiveness analyses will be performed, namely, a within-trial evaluation where cost and health effects of individual patients are limited to the 1-year follow-up period of the trial and a decision model approach where effects are modelled to incorporate longer-term impacts of the intervention.

The primary outcome of the economic evaluation is the ICER of RA compared with LA in AVF creation expressed in cost per QALY (£/QALY). All intervention resource use and access-related resource use will be derived from the secondary outcome measures and unit costs applied to all resource use estimates informed, where possible, from standard UK sources. A bottom-up approach will be used to estimate the costs associated with the two anaesthesia procedures. Effects will be captured at the individual patient level and QALYs will be derived by combining overall survival with utility weights derived from the EQ-5D questionnaire values obtained at the preoperative time, at 3 and 12 months after treatment.

A discrete-time state-transition Markov model will then be used to assess the long-term economic impact of the intervention beyond the trial period, with each cycle consisting of relevant events (ie, maturation/functional patency, failure, complications, reintervention, alternative access, adverse events (AEs) and death). Events will be driven by transition probabilities within the model,

## Box 2 Outcome measures

**Primary outcome measure**
► Unassisted functional patency of the index fistula at 1 year.

**Secondary outcome measures**
Access-specific outcome measures
► Patency (ie, is the fistula running ?): defined clinically as the presence of a bruit.
► Access complications (including infection, stenosis, thrombosis, steal and bleeding).
► Reoperation/reintervention to maintain or re-establish patency (revisional surgery, angioplasty, stenting or thrombectomy).
► Alternative accesses (eg, central venous catheters).
► Time to first cannulation.
► Cannulation difficulties (including failure to establish two-needle dialysis, infiltration and prolonged bleeding).
Patient-specific outcome measures
► Mortality
► Date commenced on HD.
► Access modality at start of HD.
► Change of RRT modality.
► Change of access modality.
► Access-related hospitalisation.
HRQoL
► EQ-5D-5L (EuroQol)[31] (crude health status measure and cost-effectiveness analysis).
► Kidney Disease Quality of Life Short Form[32] (renal-specific HRQoL).
► Vascular Access-Specific Quality Of Life[33] (vascular access-specific HRQoL).
Anaesthesia
► Pain score at incision, at 30 min and 1 hour postoperatively (NRS 0–10).
► Speed of onset/quality of motor and sensory block.[21]
► Need for anaesthetic supplementation.
► 'Failed block'.
► Volume of anaesthetic agent (mL).
► Time to administer anaesthetic (min).
Other
► Change in surgical plan, for example, switch from brachiocephalic fistula to radiocephalic fistula prior to surgical incision.

HD, haemodialysis; HRQoL, health-related quality of life; NRS, numeric rating scale; RRT, renal replacement therapy.

being informed partly by within-trial data in the short-term (ie, up to 1 year) and other sources (literature, electronic health records, etc) in the long-term (ie, beyond 1 year).

### Retention/withdrawal criteria
Participants may voluntarily withdraw from the study at any time. However, due to the nature of the intervention, it is impossible to change the allocated treatment once the anaesthetic procedure has been performed. Follow-up visits will be timed to coincide with dialysis sessions to minimise follow-up burden and to promote trial retention.

### Data collection and data management
Study specific data, which is non-identifiable, will be collected on the electronic case report form (eCRF)

using a unique patient identifier for reporting. Only the study site will have access to the identifiable information to maintain participant confidentiality. Pseudoanonymised data entered into the eCRF will be managed and stored by the RCB. The RCB systems are fully validated in accordance with industry and regulatory standards and incorporate controlled access security. Data integrity is assured by strictly controlled procedures, including secure data transfer procedures. A computer database will be constructed specifically for the trial data and will include range and logic checks to prevent erroneous data entry. Independent checking of data entry will be periodically undertaken on small subsamples. The trial statistician will also regularly check the balance of allocations by stratification variables.

All essential documents will be archived in a secure commercial vault for a minimum of 5 years after completion of the trial. Trial data will be stored under controlled conditions for at least 10 years after closure. During this period, all data will be accessible to the competent authorities and the sponsor for audit and monitoring purposes with suitable notice.

### Sample size
A total of 566 subjects (283 per arm) are required to detect a 15% difference in the primary outcome measure with 5% significance level and 90% power, assuming that 15% of subjects will be lost to follow-up, will change RRT modality or die.

Fifteen per cent is considered to be the minimum clinically important difference between the two cohorts. It is a conservative estimate of the 19% difference in 1-year unassisted functional patency observed in the results from our single-centre RCT and is the magnitude of difference considered appropriate by experts following independent review of the protocol by the UK Renal Trials Network (UKRTN).[12] UKRR data indicate that 47% of incident patients currently commence HD via an AVF/AVG.[6] A 15% increase in functional patency would allow the renal association target of 60% to be achieved.

### Statistical analysis plan
All statistical analyses will primarily be performed according to the intention-to-treat principle. However, additional analysis will be prespecified to address 'failed blocks' (eg, per-protocol, as treated, and complier-average causal effects analyses). Baseline demographics will be summarised by treatment group without formal statistical comparison. The primary outcome will be analysed using logistic regression, adjusting for stratification variables used at randomisation and the treatment group assigned. The treatment effect will be reported with a 95% CI for the OR and p value also reported. Time to loss of functional access will also be analysed using survival analysis regression methods. Similarly, for each of the secondary outcomes, analyses will be conducted using appropriate regression methods reporting the treatment effects, 95% CIs and p values. Safety data including the number of

adverse events (AEs) and serious adverse events (SAEs) will be reported overall and by study arm, where no formal statistical testing will be carried out.

## Interim analysis and early termination criteria

A 4-month internal pilot will be employed, principally to assess feasibility of recruitment. Stop–go (traffic light) criteria for continuance to the full trial will be used:

► Red: stop if <50 patients are recruited or if <5 centres are open to recruitment.
► Amber: enrol more centres if between 48 and 95 patients are recruited.
► Green: continue within existing parameters if >95 patients are recruited.

If there is failure of adherence to trial protocol in >20% of participants or significant safety concerns are raised by the IDMC, the trial will not progress beyond the pilot phase. In the event that the trial was to be terminated following the internal pilot, all patients would be followed up until the end of trial.

## Embedded process evaluation study

An embedded process evaluation study will run in parallel with the trial. The rapid feedback evaluation approach delivered by the Rapid Research Evaluation and Appraisal Lab at the Department of Targeted Intervention, University College London will combine qualitative data obtained from semistructured interviews with patients, carers and staff and documentary analysis (reports, meeting minutes, etc) to

► Explore staff views and experiences with different approaches to recruitment.
► Examine patient and carer experiences in trial participation (understanding of trial literature, experience with treatment options and reasons for withdrawal).
► Examine patient and carer experiences of declining to take part in the trial.
► Identify barriers and enablers to trial set-up, recruitment and delivery from the point of view of staff.

Data obtained will be analysed and shared with researchers throughout the main ACCess study at a time when they can be used to inform within trial decision-making processes.[26]

## AE reporting

In accordance with the Research Governance Framework for Health and Community Care, any untoward medical occurrence in a trial participant will be considered an AE, recorded in the patient's case notes and assessed for severity.[27] Any AE that is life-threatening; results in death, birth defect or significant disability; or requires hospitalisation is considered an SAE. The following trial-specific AEs will also be considered SAEs:

► A recognised perioperative complication of regional or local anaesthetic administration (including pneumothorax, inadvertent arterial puncture, inadvertent intraneural/intravascular injection, persistent neuropraxia and LA toxicity).

► The requirement for re-exploration or abandonment of surgery.

Full details including the nature of the event, start and stop dates, severity, actions taken, relationship to the trial specific intervention and outcome of all SAEs will be reported to the sponsor via the Glasgow Clinical Trials Unit on the eCRF and events will be followed up until satisfactory resolution. All SAEs will be assessed for causality and expectedness. Any SAE believed to be related to a trial-specific procedure that is thought to be unexpected (ie, the event is not listed within the protocol nor would not be expected to occur when carrying out the trial-specific procedure in normal clinical practice) will be considered a related unexpected serious adverse event (RUSAE) and must be reported to the sponsor within 24 hours of the site becoming aware. All SAEs will be reported to the IDMC, TSC and sponsor. The sponsor must inform the REC of any RUSAE within 15 days.

## Trial management and audit

The TSC, including a patient representative, will provide overall supervision of the trial and ensure that trial conduct is in line with standards set out in the EU Good Clinical Practice (GCP) Guideline.[28] The TSC (including the chief investigator, trial statistician and five independent experts) will meet on six occasions during the trial, review blinded safety data biannually and report formally to the sponsor.

An IDMC will be responsible for monitoring data emerging from the trial, in particular, as they relate to the safety of participants. The IDMC will be completely independent of the trial and any institutions involved in the trial. It will consist of an expert clinical trialist (chair), and expert in the field of vascular access and an expert statistician. The IDMC will meet annually during the recruitment and follow-up phases of the trial. The IDMC is the only body that will have access to the unblinded comparative data during the trial. Ultimate responsibility in deciding whether or not to act on recommendations from the IDMC or a decision for early termination lies with the TSC in conjunction with the sponsor and funder.

Following risk assessment, it has been determined that the study will not be routinely monitored by the sponsor; however, the sponsor randomly selects a number of studies to be audited annually. In addition, audits can be requested by individual participating sites/TSC.

## Public and patient involvement

Patients and the public have been integral to the design and implementation of this trial. Consultation and focus groups identified both the importance of access functionality, which is reflected in our choice of unassisted functional patency as the primary outcome measure, and the 'exhaustion and loss of control' experienced by patients on dialysis. The trial protocol reflects these challenges such that follow-up will, where possible, be performed while the patient is on dialysis to minimise the burden additional unnecessary hospital visits, and cannulation

diaries have been developed so that patients participating in the study will have some ownership for collecting data and reporting outcomes. The embedded process evaluation study will also explore patient and carer experiences of trial participation. A patient representative will sit on the TSC to ensure that the patient's voice is heard throughout the trial. Study participants will receive results via their dialysis units, social media, renal charities and patient groups.

## Ethics and dissemination

The trial protocol has been approved by the West of Scotland Research and Ethics Committee (REC) 3 (20/WS/0178). Research will be conducted in accordance with the Declaration of Helsinki, the Principles of GCP, the Data Protection Act (2018), the General Data Protection Regulation and the UK Policy Framework for Health and Social Care Research. Substantial amendments that require review by REC will not be implemented until the REC grants a favourable opinion for the trial (amendments may also need to be reviewed and approved by the NHS R&D departments before they can be implemented in practice at local sites).

## Consent

A member of the research team will obtain written informed consent prior to administration of any trial intervention (online supplemental appendix 2). Participants will also be asked to provide consent for future data-linkage studies via the Scottish Renal Registry and UKRR. All patients will have the right to refuse participation and withdraw from the trial at any time without providing reasons and without prejudicing further treatment.

## Access to data

The anonymised participant-level dataset and statistical code of generating the results will be made publicly available within 12 months of the end of trial via an online data repository.

## Ancillary and post-trial care

The anaesthetic (both regional and local) reflects a single-event intervention; therefore, contingency plans for the provision of ongoing treatment for individual trial participants is not required. The sponsor is a member of the Clinical Negligence and Other Risks Indemnity Scheme, which covers the sponsors legal liability in relation to clinical trials including clinical negligence and harm from study design.

## Publication/dissemination

Ownership of the data arising from this study resides with the grant holders. Research findings will be published in the name of the ACCess Collaborative Group, acknowledging the writing group as authors. Results of this trial will have implications for patients and clinicians across a range of disciplines including nephrology, anaesthesia, vascular access surgery and dialysis nursing. The principal target audience, however, is healthcare commissioners

and policy makers. Results will be published in open-access peer-reviewed journals within 12 months of completion of the trial. We will also present our findings key national/international renal meetings and support dissemination of trial outcomes directly to patients via patient groups, renal charities and social media.

**Author affiliations**
[1]Department of Anaesthesia, NHS Greater Glasgow and Clyde, Glasgow, UK
[2]Academic Unit of Anaesthesia, Critical Care and Pain Medicine, University of Glasgow, Glasgow, UK
[3]Department of Renal Surgery, Queen Elizabeth University Hospital, Glasgow, UK
[4]Institute of Cardiovascular and Medical Sciences, University of Glasgow, Glasgow, UK
[5]Department of Vascular Surgery, Queen Elizabeth University Hospital, Glasgow, UK
[6]Department of Nephrology, Queen Elizabeth University Hospital Campus, Glasgow, UK
[7]Centre for Perioperative Medicine, University College London, London, UK
[8]Anaesthesia and Critical Care, University College London Hospitals NHS Foundation Trust, London, UK
[9]Rapid Research Evaluation and Appraisal Lab, University College London, London, UK
[10]Independent Health Economist, Healthonomics UK Ltd, Reading, UK
[11]Department of Surgery, Cambridge University, Cambridge, UK
[12]Department of Surgery, Addenbrooke's Hospital, Cambridge, UK
[13]Department of Nephrology and Transplantation, Royal Free London NHS Foundation Trust, London, UK
[14]Department of Surgery and Interventional Science, University College London, London, UK
[15]Department of Transplantation, Guy's and St Thomas' NHS Foundation Trust, London, UK
[16]Department of Vascular Surgery, University Hospital Hairmyres, East Kilbride, UK
[17]Department of Anaesthesia, Freeman Hospital, Newcastle upon Tyne, UK
[18]Department of Nephrology, Dumfries and Galloway Acute Hospitals, Dumfries, UK
[19]Department of Vascular Surgery, Ninewells Hospital and Medical School, Dundee, UK
[20]Robertson Centre for Biostatistics, University of Glasgow, Glasgow, UK
[21]Department of Anaesthesia and Perioperative Medicine, Guy's and St Thomas' NHS Foundation Trust, London, UK
[22]Centre for Human and Applied Physiological Sciences, King's College London, London, UK
[23]Department of Anaesthesia, Belfast Health and Social Care Trust, Belfast, UK
[24]Department of Anaesthesia, Queen Elizabeth University Hospital, Glasgow, UK
[25]Department of Anaesthesia, Royal Free London NHS Foundation Trust, London, UK

**Acknowledgements** The authors acknowledge the contribution of Ewen Maclean, patient and public involvement representative for trial design and implementation, and Chloe Knott, patient representative on the Trial Steering Committee.

**Collaborators** Gavin Pettigrew, Regin Lagaac (Addenbrooke's Hospital, Cambridge University Hospitals NHS Foundation Trust), Rosie Hogg (Belfast City Hospital, Belfast Health and Social Care Trust), Stuart Suttie, Andrew Dalton, Samira Bell, Rose Ross (Ninewells Hospital, NHS Tayside), Thalakunte Muniraju, David McNair, Linda Stiff, Catherine Jardine (Dumfries and Galloway Royal Infirmary, NHS Dumfries & Galloway), Rita Singh, Mohammed Tariq Dosani, Jennifer Sainsbury (The Freeman Hospital, Newcastle-upon-Tyne Hospitals NHS Foundation Trust), Nikolaos Karydis, Kiran Sran, Kariem El-Boghdadly, Nadia Castrillo (Guy's and St Thomas' NHS Foundation Trust, London), James Gilbert, Sanjay Sinha, Sheera Sutherland, Sarah Crosbie, Madita Gavrila (Churchill Hospital, Oxford University Hospitals NHS Foundation Trust), Alex Vesey, Sandra Montgomery, Tina McLennan, Nina Tarkowska, Scott Oliver, Liz Brown, Shelley McLachlan (Monklands and Hairmyres Hospitals, NHS Lanarkshire), Jonathan de Siqueira, Max Troxler, Nikki Dewhirst, Mark Wright, Chetan Srinath (Leeds General Infirmary, Leeds Teaching Hospitals NHS Trust), Philip Bennett, Darren Morrow (Norfolk and Norwich University Hospitals NHS Foundation Trust Norwich), Emma Aitken, Marc Clancy, Iain Thomson, Andrew Jackson, Karen Stevenson, David Kingsmore, Margaret Aitken (Queen Elizabeth University Hospital, Glasgow, NHS Greater Glasgow and Clyde), Reza Motallebzadeh, Vishal Nangalia,

Vashist Deelchand, Rani Badhan (Royal Free London NHS Trust, London), Rajesh Sivaprakasam, Gareth Ackland Tim Egan, Matt Wikner (The Royal London Hospital, Barts Health NHS Trust), Alan Macfarlane, Rachel Kearns (Stobhill Ambulatory Care Hospital, Glasgow, NHS Greater Glasgow and Clyde), Lucian Gaianu (Independent Health Economist), Alex MacConnachie, Kirsty Wetherall, Patrick Mark (University of Glasgow), Ramani Moonsinghe, Cecilia Vindrola (University College London).

**Contributors** AJRM, RJK, MJC, DK, KS, AJ, PM, MA, RM, CV-P, LG, GP, RM, NK, AV, RS, TM, SS, AMcC, KW, KE-B, RH, IT and EA contributed to the development and implementation of this protocol and approved the manuscript.

**Funding** This work is supported by National Institute for Health Research (NIHR) (Health Technology Assessment 130 567). The views expressed are those of the authors and not necessarily those of the NIHR or the Department of Health and Social Care.

**Competing interests** AJRM is currently president of Regional Anaesthesia UK (RA-UK). RA-UK has reviewed and endorsed the trial protocol, but has not been involved in the design or development in any way. AJRM has received consulting fees from Heron therapeutics and Intelligent US. RH has received honoraria from GE.

**Patient consent for publication** Not applicable.

**Provenance and peer review** Not commissioned; externally peer reviewed.

**ORCID iDs**
Alan JR Macfarlane http://orcid.org/0000-0003-3858-6468
Cecilia Vindrola-Padros http://orcid.org/0000-0001-7859-1646
Alex McConnachie http://orcid.org/0000-0002-7262-7000
Kariem El-Boghdadly http://orcid.org/0000-0002-9912-717X

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
