## [Reviewer comments · BMJ Open]

ARTICLE DETAILS

TITLE (PROVISIONAL)	Anaesthesia Choice for Creation of arteriovenous fistula (ACCESS) study protocol : a randomised controlled trial comparing primary unassisted patency at one year of primary arteriovenous fistulae created under regional compared to local anaesthesia
AUTHORS	Macfarlane, Alan; Kearns, Rachel J; Clancy, Marc James; Kingsmore, David; Stevenson, Karen; Jackson, Andrew; Mark, Patrick; Aitken, Margaret; Moonsinghe, Ramani; Vindrola-Padros, Cecilia; Gaiuanu, Lucian; Pettigrew, Gavin; Motallebzadeh, Reza; Karydis, Nikolaos; Vesey, Alex; Singh, Rita; Muniraju, Thalakunte; Suttie, Stuart; McConnachie, Alex; Wetherall, Kirsty; El-Boghdadly, Kariem; Hogg, Rosemary; Thomson, Iain; Nangalia, Vishal; Aitken, Emma

VERSION 1 – REVIEW

REVIEWER	Prasannarong, Mujalin Chiang Mai University Faculty of Associated Medical Sciences, Physical Therapy
REVIEW RETURNED	16-Jun-2021

GENERAL COMMENTS	This manuscript is an RCT protocol that compares the effects of RA and LA on one-year primary patency in AVF patients. This study is very interesting and the methods seem to be well planned. However, some information should be included in the manuscript. I hope that all suggestions and questions may be useful for the authors. 1. The title should be concise and please add “protocol” in the title.2. It would be great if the authors rewrite the background in the introduction part to encourage the rationale of the study and lead to a specific objective of the experiment. The rationale for measuring secondary parameters may not quite clear. More information about the relationship between treatments and outcome measures is needed.3. One-year primary patency is a very interesting primary parameter. Is it possible if the author compares the effect of RA and LA on AVF maturation parameters? For example, maturation number and maturation rate.4. Although a multicenter study may be a strength of this experiment, the authors need to concern about confounding factors due to multicenter study settings. The author should mention and propose a way to control the confounding factors in the manuscript.
--

	5. The authors mentioned adjusting for stratification variables, please add possible variables in the manuscript. 6. Are there any subgroup analyzes of this study? To gain more reliable results, please list, define, and explain. 7. Some secondary outcomes may not be directly affected by the treatment (for example, mortality, access modality at start of HD, and change in surgical plan). The secondary outcomes should be reconsidered to be specific to rationales and study questions. 8. The quality of life questionnaire should be specific to the objective of the study. The authors should explain why this study needs all three questionnaires? 9. Tables (boxes) should present the key contents of the study. If the authors would like to present outcomes, the author should include the primary outcome in the outcome table (Box2). In addition, inclusion criteria should be included in the criteria table (Box1). 10. To avoid bias, the randomization process should be block randomization. 11. How the authors positive about the standardization of vascular access, and anesthetic techniques among surgeons and anesthetists in a different setting? 12. According to multicenter study settings, how the authors manage sample size in each setting? 13. The authors should include the discussion part in the manuscript to share their point of view and critical thinking about this project. 14. To gain a reproducible method of anesthesia, details in appendix 1 should be included in the main manuscript.
--	--

REVIEWER	Ashby, Damien Imperial College London, Renal Medicine
REVIEW RETURNED	06-Sep-2021

GENERAL COMMENTS	This is a well-designed study, which should give us high quality answers, relevant to a very important clinical problem: arteriovenous fistula primary failure. I have one major criticism and a number of minor comments, but none are sufficient to reduce the overall scientific quality and clinical relevance of the study. Major 1. The economic evaluation. The within trial evaluation is entirely appropriate, but the decision model is an extrapolation study, which takes place beyond the RCT. Moreover, the Markov model is a complex method which needs specialised peer review. I think this should be presented as a separate study, because (1) in reality it is a separate study (with methods and concepts very different from the RCT) and (2) otherwise the conclusions of the modelling study may be accepted as if they are of the same quality as the conclusions of the RCT. It feels confusing to be planning
---

	this study within the same protocol, or reporting it within the same paper. Minor 2. Observer blinding will be hard to achieve, since the patients, with whom staff have a relationship, are not blinded. Not sure what can be done about this however, and I don't think it's too serious since the main outcomes are quite objective. 3. The title should perhaps make it more clear that this is a protocol publication. 4. In a few places claims are oversimplified to a misleading extent (though none of these is serious or sufficient to undermine the arguments supporting the study). As examples: (a) Page 7 line 30. This is a common misstatement which glosses over the bias of observational studies. Suggest emphasising that the AVF needs to be successful (and therefore that this statement is not describing 'intention to treat' analyses), for example: "...when comparing dialysis after successful AVF formation with dialysis via..." (b) Page 17, line 30. This statement appears to suggest that a 15% improvement in functional patency would lead to a 15% improvement in fistula prevalence in the dialysis population. But fistula prevalence is complex, depending on many factors, and one cannot draw this conclusion.
--	---

VERSION 1 – AUTHOR RESPONSE

Reviewer: 1

1. The title should be concise and please add "protocol" in the title.

"Protocol" added as requested. We have shortened the title a little but as per the SPIRIT statement we felt we must still provide sufficient detail of the design, population and intervention.

2. It would be great if the authors rewrite the background in the introduction part to encourage the rationale of the study and lead to a specific objective of the experiment. The rationale for measuring secondary parameters may not quite clear. More information about the relationship between treatments and outcome measures is needed.

Modifications have been made to the introduction to reflect both reviewers' comments, improve clarity and emphasise the specific rationale for the study.

The primary objective of the study is to determine whether or not the vasodilatation, which results from sympathetic blockade observed with regional anaesthesia, translates into clinical improvements in functional fistula patency.

Secondary outcome measures have been chosen to better reflect the overall burden of care on the patient; describe access-related interventions necessary to achieve functional patency; and facilitate cost-effectiveness analysis. This is discussed in greater detail elsewhere in the manuscript and responses.

3. One-year primary patency is a very interesting primary parameter. Is it possible if the author compares the effect of RA and LA on AVF maturation parameters? For example, maturation number and maturation rate.

Historically the terminology around vascular access patency is confusing, not patient-focussed and open to misinterpretation. The term 'primary patency' is commonly used to mean a fistula that maintains blood flow without intervention irrespective of whether or not it is suitable to sustain dialysis; 'primary assisted patency' refers to a fistula that needs an intervention to maintain primary patency; while 'secondary patency' is used to describe an access that has temporarily thrombosed, but in which it has been possible to re-establish patency (Huijbregts et al, 2008). None of these terms convey whether or not the fistula can actually be used for dialysis. As such we have deliberately avoided using them anywhere within the protocol, instead opting for measures of 'function' and 'additional interventions needed to maintain patency'. As such, all the traditional definitions will be possible to quantify, but additionally, we will be better able to define the burden of a suboptimal fistula or access.

Similarly the term 'maturation' is open to interpretation. We presume the reviewer is using it to mean 'functional patency' i.e. whether or not a fistula is useable for dialysis. Maturation is very subjective, especially in the pre-dialysis population, depending on local cannulation practices, timing of dialysis initiation, experience of dialysis nurses etc.

The reviewer comments that the primary outcome measure for this study is "one-year primary patency". This is not the case. The primary outcome measure, as stated on page 11, is "one-year unassisted functional patency". This primary outcome measure was chosen after extensive consultation with PPI groups and aligns closely with the SONG-HD consortium consensus on standardised reporting outcomes for vascular access (Viecelli et al, 2018). It was chosen because it reflects both the need for an access to adequately provide dialysis (i.e. "there's no point having a fistula if it doesn't work") and 'uninterruptedly' sustain dialysis without need for intervention, as the burden to patients of repeated interventions to facilitate maturation or maintain function is significant. Within this clarified, we believe the primary outcome measure that we've chosen does capture all the data that the reviewer is interested in regarding 'maturation'. Finally, rather than using the term primary objective on page 6 in the first paragraph of the methods where we also described unassisted functional AVF patency we have changed this to primary outcome to make this clearer.

4. Although a multicenter study may be a strength of this experiment, the authors need to concern about confounding factors due to multicenter study settings. The author should mention and propose a way to control the confounding factors in the manuscript.

Having previously conducted a single-centre study with a similar protocol (Aitken et al, 2016), both the corresponding and lead author anticipate that the multicentre nature of this work will address many of the criticisms regarding generalisability and confounders highlighted for our previous single centre study (namely a single regional anaesthesia 'expert' performing all the blocks, perceived high early failure rates in both arms of the study, underpowered for subgroup analysis etc). The current study was designed following an NIHR research call for a multicentre study to address exactly these issues.

It is acknowledged that both anaesthetic and surgical practice varies between centres and there will be variable levels of skill/experience amongst clinicians/ operators. To control for this, both anaesthetic and surgical technique has been standardised (as outlined on page 8 and 10 of the manuscript respectively). Only centres that already offer both anaesthetic techniques (LA and RA) as 'standard practice' will be eligible to participate in the study, so we don't capture any 'learning curve' effect.

Peri-operative care will inevitably vary between units e.g. use of a block room, fasting guidance, same day discharge. We have taken a pragmatic approach to this and, instead of standardising completely (the anxiety was that this might hinder recruitment), data will be captured on all of these variables, both to provide a 'snapshot' of current practice around the country and to facilitate the cost-effectiveness analysis.

Patients will be recruited from a range of high (>150 fistula per year) and medium (50-150 fistula per year) centres to ensure that the outcomes from the study will be generalisable. Randomisation will take place at a patient, not centre, level to minimise bias from inevitable small variations in surgical and dialysis practice.

5. The authors mentioned adjusting for stratification variables, please add possible variables in the manuscript.

Details of the stratification variables (centre, dialysis status and access site) are outlined on page 7.

6. Are there any subgroup analyzes of this study? To gain more reliable results, please list, define, and explain.

The study is not powered for subgroup analysis. We arrived at this decision after careful consideration of the relative merits versus need for a significantly larger sample size and the ability to achievably recruit within a reasonable timescale, concluding that it was more important to ensure we delivered on the primary aim than facilitate subgroup analysis. Nevertheless, data from our previous single centre study (Aitken et al, 2016) suggested that almost all of the improvements in AVF patency with block were observed in RCF. Therefore the statistical analysis plan does allow data for RCF and BCF to be explored and analysed separately.

7. Some secondary outcomes may not be directly affected by the treatment (for example, mortality, access modality at start of HD, and change in surgical plan). The secondary outcomes should be reconsidered to be specific to rationales and study questions.

The secondary outcome measures (including mortality and dialysis modality) form part of the "standard CKD set" recommended by the International Consortium for Health Outcomes Measurement (ICHOM) CKD Working Group and the SONG-HD consortium and reflect the minimum dataset to be collected in all studies of haemodialysis patients (page 11).

Whether or not mortality is affected by vascular access is a matter of debate, with many large observational studies demonstrating higher mortality in patients dialysing via catheters (Malas et al, 2015; Yeh et al, 2019). Therefore we do consider mortality to be directly relevant to the study rationale (if the fistula doesn't achieve functional patency, alternative accesses, with associated higher infection and mortality rates, will be required).

As outlined in box 2, secondary outcome measures have been chosen to reflect the overall patient as well as the vascular access specifically. Additionally anaesthetic specific and safety end points are included. An additional statement has been added to the manuscript on page 11 to detail this further.

The decision to include "change of surgical plan" as a secondary endpoint reflects an observation made from our single centre study that, in patients receiving RA, the vasodilatation observed following block led to a change in the operative plan with more distal AVF created. Other case series report similar results (Laskowski et al. *Ann Vasc Surg* 2007; 21: 730-3). Given the well-recognised benefits of RCF over BCF in terms of long-term durability and preservation of venous capital, we believe that this is a valid secondary endpoint, previously demonstrated to be affected by the intervention.

The range of secondary endpoints has been chosen with the intention of capturing the entire patient experience resulting from suboptimal/failed fistula creation (the implications of not having a mature fistula, need to commence dialysis via a line, complications and interventions on the same). We robustly defend our choice of secondary endpoints as they both allow us to accurately capture the patient journey, including negative life events resulting from suboptimal vascular access, and are essential to perform cost-effectiveness analysis.

8. The quality of life questionnaire should be specific to the objective of the study. The authors should explain why this study needs all three questionnaires?

EQ-5D is a health status tool, utilised by the National Institute for Health and Care Excellence (NICE), to make most funding decisions. It is very crude in its assessment of overall health status, but if the results of this trial are to influence future practice and commissioning decisions, its inclusion is essential for cost-effectiveness analysis purposes.

KD-QOL is a more detailed quality of life tool, validated specifically for patients with end-stage renal disease. We anticipate that it will detect more subtle changes in HR-QOL than EQ-5D, especially in relation to renal failure symptomatology.

Finally, VASQOL is recently validated (Richarz et al, 2021) HR-QOL tool that specifically evaluates the impact of the vascular access on health status (e.g. it captures the anxiety induced by poorly functioning access etc).

By employing three different HR-QOL tools we hope to capture not just gross differences in HR-QOL but also renal and access specific differences. These subtleties, especially in relation to vascular access, often prove difficult to capture, as the overall dialysis status often dominates traditional HR-QOL measures.

We agree with the reviewer that this detailed explanation of our choices would benefit from being included within the main protocol text, however we are limited by word count. Therefore, we've added a short note on the justification for each HR-QOL measure to box 2.

9. Tables (boxes) should present the key contents of the study. If the authors would like to present outcomes, the author should include the primary outcome in the outcome table (Box2). In addition, inclusion criteria should be included in the criteria table (Box1).

As requested, inclusion criteria have been added to box 1 and the primary outcome measure to box 2. The manuscript has also been adjusted to reflect this (page 7)

10. To avoid bias, the randomization process should be block randomization.

A method of permuted blocks will be used as described on page 7.

11. How are the authors positive about the standardization of vascular access, and anesthetic techniques among surgeons and anesthetists in a different setting?

Like all randomised trials of procedural/ surgical technique, complete standardisation of the operative/ anaesthetic technique is difficult to definitively ensure. Similarly, we've been cognisant that too prescriptive a protocol may hinder recruitment. We've therefore standardised key aspects of both anaesthetic (drug mix, volume, procedure if initial block suboptimal) and surgical (arteriotomy size, anatomy, suturing technique), leaving some flexibility for individual clinicians to adapt to the specific

situations they may face. These 'key elements' are outlined on pages 8-9 (anaesthetic) and 10 (surgery) of the manuscript respectively.

Only centres that create >50 fistula per year will be eligible to participate. As previously highlighted, only centres which already offer both arms of the study as routine care will be eligible to participate, thus eliminating the potential confounding effects of a 'learning curve' for one or other technique. All surgery and anaesthesia must be performed either by the consultant or by a trainee under direct consultant supervision. Protocol deviations and adverse events stratified by centre will also be regularly be monitored by the IDMC committee.

12. According to multicenter study settings, how the authors manage sample size in each setting?

We are unsure exactly what is meant by this question, however believe it relates to differential recruitment across centres. If this is not the case please let us know and we'd be happy to address any other issues.

It's well recognised that most multicentre studies have differential recruitment between centres, with the co-ordinating site often recruiting more patients than other sites. As previously mentioned and described on page 6 of the manuscript, patients will be recruited from a mixture of high (>150 AVF/ yr) and medium (50-150 AVF/yr) volume centres. Any centre that has not recruited a patient after 3 months will be closed and substituted with an additional site.

Furthermore, two measures will be employed to minimise bias created by potential variation in outcomes between centres:

- a. Randomisation is stratified by centre
- b. The unblinded IDMC will monitor protocol adherence and adverse events by site, so any site with significantly outlying outcomes or recruitment can be identified and addressed within the trial.

13. The authors should include the discussion part in the manuscript to share their point of view and critical thinking about this project.

We welcome the opportunity to respond to the reviewers' comments, share our critical thinking and rationale for the trial design. We have followed the journals recommended format for trial protocol manuscripts. This does not include a discursive section. The 4,000 word limit also precludes additional discussion within the main manuscript.

14. To gain a reproducible method of anaesthesia, details in appendix 1 should be included in the main manuscript.

Given we're limited by the word count for the main manuscript, additional details of the RA technique have been included within Appendix 1. At the reviewer's request, additional details of minimum anaesthetic volumes have been moved to page 8 of the main manuscript.

Reviewer: 2

1. The economic evaluation. The within trial evaluation is entirely appropriate, but the decision model is an extrapolation study, which takes place beyond the RCT. Moreover, the Markov model is a complex method which needs specialised peer review. I think this should be presented as a separate study, because (1) in reality it is a separate study (with methods and concepts very different from the RCT) and (2) otherwise the conclusions of the modelling study may be accepted as if they are of the same quality as the conclusions of the RCT. It feels confusing to be planning this study within the same protocol, or reporting it within the same paper.

We agree wholeheartedly with the reviewer that the health economic (HE) evaluation requires separate consideration. We have discussed this extensively with Dr Gaianu, health economist, and a separate more detailed health economic methodology protocol paper is being prepared. Nevertheless, given the original brief for this study and in response to reviewer 1's comments requesting justification for the secondary endpoints (many of which are necessary for the HE evaluation), we feel it's necessary to retain some discussion of the HE aspects of the protocol within this primary protocol text as rationale for the choice of end points and planned outputs and dissemination of trial findings upon completion i.e. the intention would be to make a case for or against RA to commissioners.

We have however shortened the section on HE evaluation to focus more on the in trial analysis rather than the extrapolatory Markov modelling.

Minor

2. Observer blinding will be hard to achieve, since the patients, with whom staff have a relationship, are not blinded. Not sure what can be done about this however, and I don't think it's too serious since the main outcomes are quite objective.

We agree, as outlined at the bottom of page 10, that it is impossible to blind patients without sham procedure. This risks unblinding investigators also. We did consider the possibility of sham RA but after extensive discussion and PPI consultation during the protocol development phase, it was concluded that the risk of complication (pneumothorax, vascular puncture etc) with sham procedure was not justifiable.

As the reviewer indicates, by utilising ultrasound in addition to clinical assessment of maturation, we hope to minimise observer bias and provide an objective measure of functional patency.

3. The title should perhaps make it more clear that this is a protocol publication.

This has been modified.

4. In a few places claims are oversimplified to a misleading extent (though none of these is serious or sufficient to undermine the arguments supporting the study). As examples:

(a) Page 7 line 30. This is a common misstatement which glosses over the bias of observational studies. Suggest emphasising that the AVF needs to be successful (and therefore that this statement is not describing 'intention to treat' analyses), for example: "...when comparing dialysis after successful AVF formation with dialysis via..."

We have refined some of the statements in the introduction to reflect the reviewer's comments including the statement, "Furthermore, there is observational data demonstrating superior survival in patients successfully dialysing via AVF compared to those using central venous catheters (CVCs) or prosthetic arteriovenous grafts (AVGs) for dialysis" on page 5.

(b) Page 17, line 30. This statement appears to suggest that a 15% improvement in functional patency would lead to a 15% improvement in fistula prevalence in the dialysis population. But fistula prevalence is complex, depending on many factors, and one cannot draw this conclusion.

On review, we agree with this comment and have removed the statement about fistula prevalence rates from the manuscript. We believe that the preceding statement regarding incident access rates is valid as there are fewer additional influences and cofounders in this situation.

Thank you again for taking the time to review our manuscript. We hope that the above has sufficiently answered the comments of both the reviewers and the editor.

VERSION 2 – REVIEW

REVIEWER	Ashby, Damien Imperial College London, Renal Medicine
REVIEW RETURNED	19-Sep-2021
GENERAL COMMENTS	Totally reasonable responses, no further comments.